# Cerebral cortical activation and muscle performance during blood flow restriction training after ischemic stroke: A randomised functional near-infrared spectroscopy study

Xinyu Bai[1,2,3☉‡], Yunyun Zhang[2,3,4☉‡], Weiwei Cao[2,3,4], Yang Xie[2,3,4], Yonggang Zhu[2,3,4*]

**1** Acupuncture and Massage Department, The Affiliated Lianyungang Hospital of Xuzhou Medical University, Lianyungang, Jiangsu, China, **2** The First Affiliated Hospital of Kangda College of Nanjing Medical University, Lianyungang, Jiangsu, China, **3** Lianyungang Clinical College of Nanjing Medical University, Lianyungang, Jiangsu, China, **4** Department of NeuroRehabilitation, The Affiliated Lianyungang Hospital of Xuzhou Medical University, Lianyungang, Jiangsu, China

☉ These authors contributed equally to this work.
‡ These authors share first authorship on this work.
* lygzhuyg@163.com

## Abstract

### Aims

To compare the effects of low-load (LL) blood flow restriction (BFR) and high-load (HL) training on cortical activation and the specific contributions of individual brain regions to functional recovery in stroke patients.

### Methods

Sixty-six patients with ischemic stroke were divided into BFR (30% one-repetition maximum [1RM]), matched LL, or HL (80% 1RM) groups. Patients underwent a four-week supervised cycling program, and oxyhemoglobin (HbO) concentrations were assessed during the first session and after the program via functional near-infrared spectroscopy (fNIRS). Muscle performance was characterized by the rectus femoris muscle cross-sectional area (CSA), knee extensor peak torque (PT), and Fugl–Meyer lower extremity (FMLE) scores.

### Results

Compared with the LL group, the BFR and HL groups presented significant brain activation (increased HbO concentration) during the first session ($P < 0.05$). Following the 4-week intervention, the BFR and HL groups presented greater changes in the HbO concentration (ΔHbO), PT and FMLE scores than did the LL group ($P < 0.05$). The ΔHbO values in the primary motor cortex (M1), premotor cortex and supplementary motor area (PMC-SMA) of the affected hemisphere (AH) were considerably greater

**Data availability statement:** All relevant data are within the manuscript and its Supporting Information files.

**Funding:** This research was supported by the Ageing Health Research Project of Lianyungang (grant numbers: L202425).The funders had no role in study design, data collection and analysis, decision to publish.

**Competing interests:** The authors have declared that no competing interests exist.

than those in the unaffected hemisphere ($P < 0.05$), whereas there was no difference in the dorsolateral prefrontal cortex (DLPFC). Changes in PT (mean $r = 0.51$ [range = 0.46–0.55]; $P < 0.05$) and FMLE scores (mean $r = 0.54$ [range = 0.48–0.62]; $P < 0.05$) were positively correlated with the AH M1 and PMC-SMA ΔHbO across groups.

## Conclusions and Implications

By actively manipulating the M1 and PMC-SMA, LL-BFR and HL training yield comparable short-term improvements in central and peripheral performance after stroke. (Registry: Chinese Clinical Trial Registry; ChiCTR2400087378).

## Introduction

Motor rehabilitation critically impacts the clinical evolution of the poststroke phase, enabling functional recovery of the affected body segment and enhancing neuroplasticity [1,2]. While various forms of resistance training (RT) demonstrate benefits in muscle hypertrophy and strength for stroke patients [3], their neurogenic mechanisms remain poorly characterized. High-load (HL) RT with 70%−85% workload at 1 repetition maximum (1RM) considered optimal for the development of further neuroadaptation, strength and muscle hypertrophy in healthy adults [4], yet its applicability to stroke populations, who often tolerate lower mechanical loads, remains contentious. Blood flow restriction training (BFRT), a novel RT approach combining partial vascular occlusion with low-load exercises (20%−30% 1RM), offers comparable muscular benefits to HLRT while minimizing joint stress in neurologically impaired individuals [5,6]. However, current BFRT research remains disproportionately focused on peripheral adaptations, leaving two critical gaps: (i) the cortical signatures of BFRT-induced neuroplasticity in stroke, and (ii) the correlation between central neural drive and functional recovery.

Neuroimaging advances now permit direct interrogation of exercise-induced cortical dynamics. Prior work in healthy populations links RT intensity to increased cortical blood flow [7,8], electroencephalography (EEG) spectral power [9], and fMRI-based blood oxygenation changes [10], with transcranial magnetic stimulation (TMS) further implicating corticospinal excitability enhancements [11]. Crucially, BFRT demonstrates amplified cortical responses in healthy individuals, with TMS showing significantly greater corticospinal excitability during upper- or lower-limb BFRT versus low load (LL) RT [12,13]. Specifically, functional near-infrared spectroscopy (fNIRS) reveals optimal motor cortex activation during squatting with moderate-pressure BFRT (250 mmHg), outperforming both non-restricted (0 mmHg) and alternative pressure conditions (150 and 300 mmHg) [7]. Yet, these findings exhibit limited generalizability to stroke populations, whose neurovascular coupling is disrupted by cerebrovascular pathology. Oxygen metabolism and hemodynamics during RT-induced neuronal activity remain understudied in neurological disorders, with existing stroke RT studies predominantly rely on pre-post cross-sectional designs, failing to capture

real-time neurodynamics that may dictate treatment responsiveness. fNIRS emerges as a pragmatic solution, enabling continuous monitoring of hemodynamic responses during rehabilitation tasks, and earlier research has established its utility for stroke exercise applications [14,15].

To resolve the critical knowledge gaps in BFRT-induced neuroplasticity, we conducted a randomized controlled trial integrating real-time fNIRS with muscle strength-structure-function profiling. We aimed to: (a) Compare acute-phase cortical activation (prefrontal-motor oxyhemoglobin[HbO]) across BFRT (30%1RM+occlusion), HLRT (80%1RM), and LLRT (30%1RM); (b) Investigate temporal associations between HbO changes and improvements in muscle strength, hypertrophy, and motor function; (c) Validate if BFRT induces HLRT-equivalent neuroplasticity with 50% lower mechanical loads, prioritizing central drive over peripheral adaptations.

## Materials and methods

### Study design and Participants

This prospective, double-blind randomized clinical trial was conducted from December 11, 2023 to August 13, 2024 in the Department of Neurorehabilitation of the First People's Hospital of Lianyungang, China. We recruited hospitalized patients aged 45–75 years who (1) suffered the first subcortical ischemic stroke attack with hemiparesis occurring within 2weeks to 6 months after the attack; (2) had a unilateral supratentorial lesion; (3) had ≥grade 3 muscle strength (manual muscle test) of the major flexor and extensor groups of the affected lower extremity; and (4) were able to cooperate in completing the fNIRS task. The exclusion criteria for patients were as follows: (1) severe cognitive dysfunction with a score of <20 on the minimal mental state examination (MMSE); (2) large cortical lesions (>1/3 of the middle cerebral artery region), severe carotid arteries (>90%), or cerebral stenosis (>75%); (3) generalized instability; and (4) severe scalp dermatitis.

The study was conducted in accordance with the principles of the Helsinki Declaration and was approved by the institutional ethics committee [KY-20231012001–02]. All patients signed an informed consent form before participation. Due to an oversight, the trial was retrospectively registered one month after initiation (Chinese Clinical Trial Registry, ChiCTR2400087378), and the study is reported in accordance with the CONSORT guidelines, with corresponding flow diagram submission as Fig 1.

### Randomization and blinding

At the initial visit, informed consent was obtained, eligibility was confirmed, and patients completed the 1RM test, during which patients were instructed to pedal, and the resistance was increased until they were no longer able to pedal. The process was repeated 5 times, and the greatest value obtained was deemed the 1RM. Computer-based stratified randomization (1:1) was generated by the statistician, and participants were stratified by sex. The resistance values recorded during the 1RM test (3–8, 9–14, and 15–20) were used to stratify the subjects into the BFR, LL, or HL groups, respectively. The statistician was informed about group assignment, but the investigator and subjects were blinded.

### Exercise protocols

MOTOmed intelligent exercise training (MOTOmed Viva 2, RECK Company, Germany) was applied in resistance mode with a cadence of 50 rpm. Seated exercise programs allow severely impaired stroke survivors to complete an exercise program without balance and safety restrictions. The BFR and LL groups (30% 1RM training load) and the HL group (80% 1RM training load) were subjected to a 3-min warm-up and 20 min (4 min [3 min pedaling + 1 min rest] as a set *5) of resistance pedaling, 5 times/week for 4 weeks. The BFR group performed the same workout as the LL group but with the addition of pneumatic cuffs (B STRONG, USA) with a width of 7 cm in the proximal portion (inguinal crease) of the affected lower extremity. Previous studies in healthy subjects have indicated that a pressure of 250 mmHg is optimal for achieving the greatest degree of cerebral activation [7]. Considering that circulation may be impaired in stroke patients, we set the

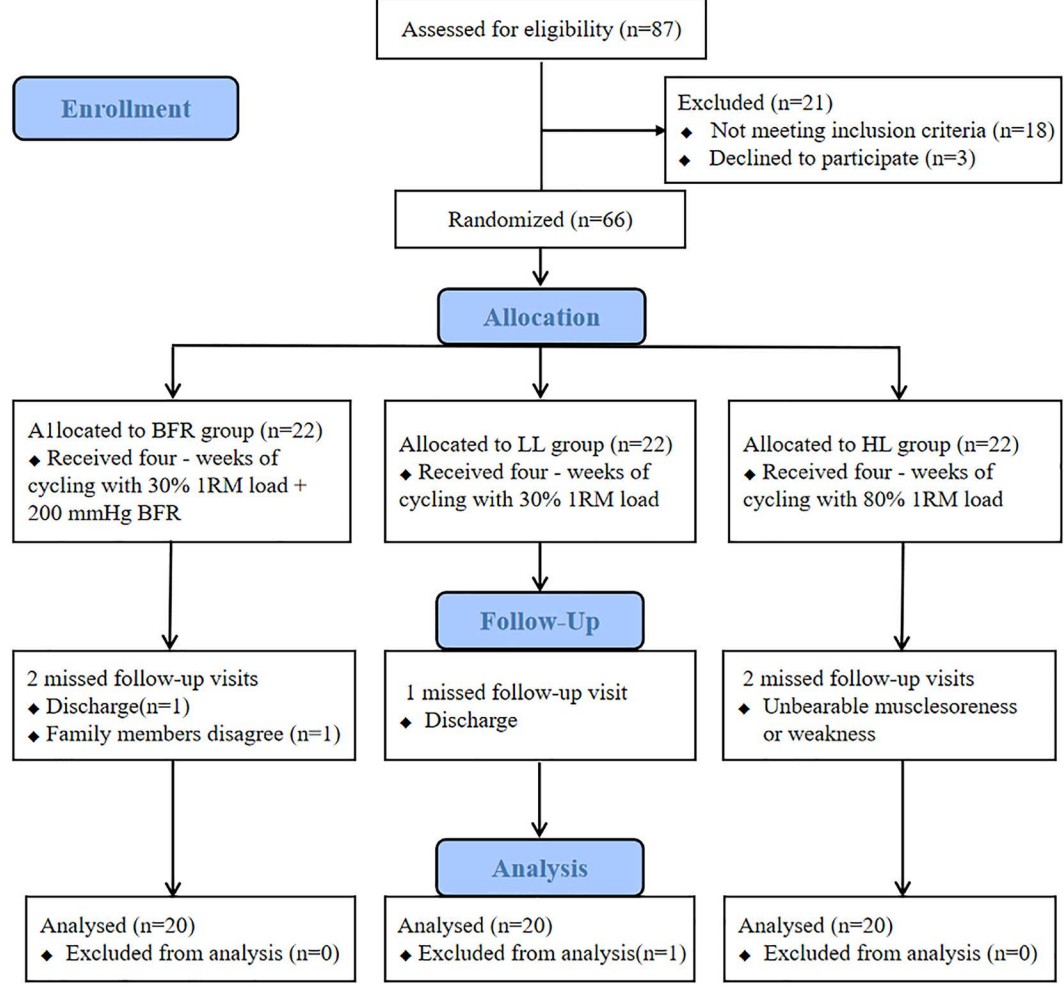

**Fig 1. Flow diagram.** BFR, blood flow restriction; LL, low load; HL, high load.

pressure value at 80% (200 mmHg). We inflated the cuffs 10 s before starting the training session, deflated the cuffs at the end of each set, and then inflated the cuffs again for the last 10 s of the inter-set rest. Notably, we used intermittent pressurization, with 3 min of pressurization during pedaling and 1 min of pressure release per set, which was repeated 5 times. In addition, all the subjects received conventional medication, physiotherapy and occupational therapy.

## Assessments

**Cortical activation.** A multichannel fNIRS device (model NirSmart-6000, Wistron, Danyang, China) was utilized to capture brain hemodynamic signals at an 11-Hz sampling rate and calculate the corresponding HbO and deoxyhemoglobin (HHb) concentrations. In subsequent analyses, HbO levels were used to infer cortical activation, mainly due to the higher signal–to–noise ratio and better measurement confidence with HbO than with HHb [16]. A total of 35 channels were positioned on the left and right dorsolateral prefrontal cortex (LDLPFC/RDLPFC), primary motor cortex (LM1/RM1), premotor cortex and supplementary motor area (LPMC-SMA/RPMC-SMA) in accordance with the International 10–20 system [17] (Fig 2a) because these areas are not only closely related to motor planning, execution,

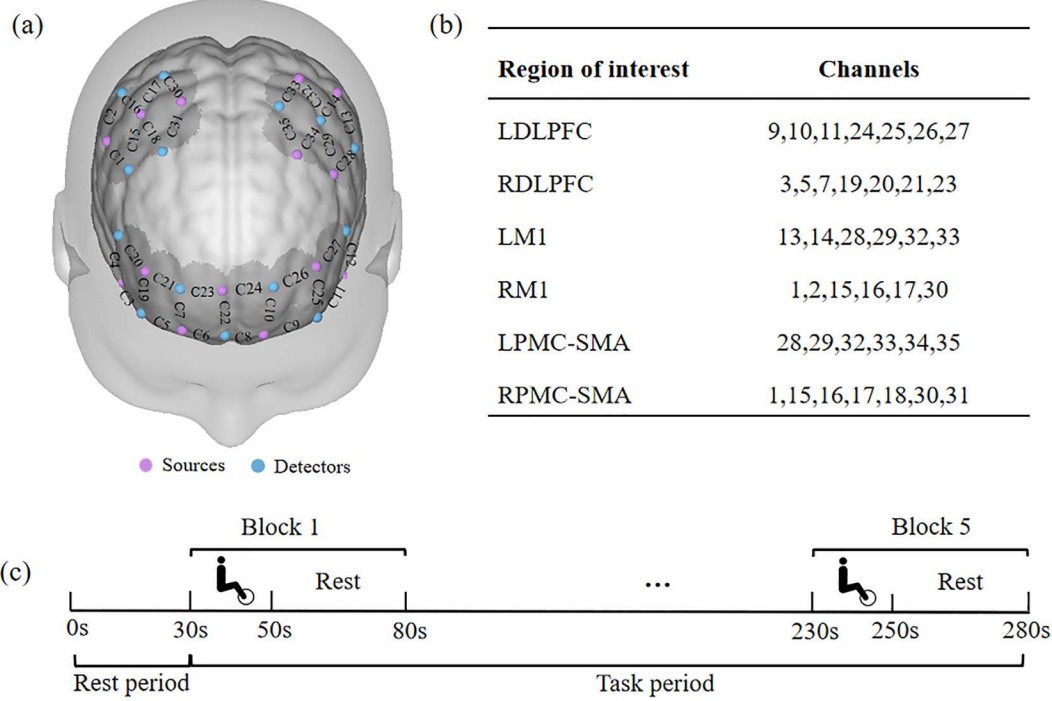

**Fig 2. (a) fNIRS brain channel locations and (b) regions of interest on the Brodmann template; (c) block design paradigm.** L/RDLPFC, left/right dorsolateral prefrontal cortex; L/RM1, left/right primary motor cortex; L/RPMA-SMC, left/right premotor and supplementary motor cortex.

and control [18], but also have been used in previous studies [19]. The Montreal Neurological Institute and Hospital (MNI) coordinates of each channel were calculated via Neural Functional Region Localization Functions [20]. Moreover, the sum of the HbO values of the corresponding channels was used for analysis (Fig 2b).

All fNIRS tests were performed on the MOTOmed. The investigator communicated the fNIRS assessment process to the participants in advance (Fig 2c). The participants remained awake, sat quietly in front of the MOTOmed, and avoided unnecessary body movements. A block design was adopted (rest [30 s], 5 cycles of pedaling [20 s], rest [30 s]). Prior to the assessment, the investigator set resistance levels for each participant according to the resistance values predetermined by the statistician (30% or 80% 1RM), with or without the use of a BFR band (see below for a detailed description) depending on the group, and the pressure of the band was released during the rest periods. All the participants practiced the task for 3–5 minutes before the assessment. fNIRS data were collected during the cycling sessions, and the subjects underwent two fNIRS scanning tests: one during the first session ($HbO_1$) and the other four weeks later ($HbO_2$). ΔHbO was calculated as the difference between the results of the two tests ($HbO_2 - HbO_1$).

## Muscle performance

Muscle morphology was determined via musculoskeletal ultrasonography (Konica Minolta, China) via an L18-4 line array probe at the knee joint location. The patient was positioned lying down, with a roller positioned under the knee pad to bend the knee at approximately 30°. The short axis of the probe was held vertically against the skin surface to measure the cross-sectional area (CSA) of the rectus femoris muscle (RF) at the upper pole of the patella of the affected leg, 10 cm above the knee. To minimize compression of the soft tissues, more coupling agent was used, and the pressure was reduced [21]. The image was then frozen, and the CSA was manually sketched ($cm^2$).

The peak torque (PT) of the extended knee on the affected side was measured via isokinetic muscular strength testing equipment (Yikang, China). Patients remained seated and were fitted with an isokinetic knee accessory. The patients completed five sets of maximal knee extension and knee flexion centripetal motions at an angular velocity of 60°/s each. The greatest muscular force of a muscle contraction was represented by the maximum peak PT of knee extension, which was measured in N.m.

The Fugl–Meyer lower extremity (FMLE) test was used to assess motor function, which was divided into five categories: supine lower extremity reflexes, flexor comovements, and extensor resistances; joint comovements in the seated position; separations in the standing position; normal reflexes and coordination/speed in the seated position; normal reflexes in the sitting position; and coordination/speed in the supine position. There were seven major items and 34 subitems, each of which received one point, for a total of 34 points. All muscle performance outcomes were evaluated before and after the intervention by the investigators.

## Data preprocessing

**fNIRS data preprocessing and analysis.** The NirSpark software package, run in MATLAB (MathWorks, USA), was used for analysis. Data preprocessing was divided into six steps: eliminating the time intervals unrelated to the experiment; removing the artifacts unrelated to the experimental data; converting light intensity to optical density; applying a bandpass filter (0.01–0.2 Hz) to filter out noise and interference signals; converting optical density into blood oxygen concentration; and setting the hemodynamic response function with an initial time of −2 s and an end time of 50 s (with "-2–0 s" as the reserved baseline state and "0–50 s" as the time for a single block paradigm). Additionally, the pedaling duration was set to 20 s, and the blood oxygen concentrations of the 5 block paradigms were superimposed and averaged to generate a block average result. An image left–right flip was also performed, i.e., the image results for the affected hemisphere (AH) were all displayed on the right side.

## Statistical analysis

On the basis of previous studies that involved the use of fNIRS to detect cortical reorganization in stroke patients [22], we used SPSS 15.0 for sample size estimation. We set an α level of 0.05 and a β of 0.10 (power level of 0.90). According to the analysis, at least 57 patients were needed. Assuming a dropout rate of 15% due to excessive movement artifacts in the exercise training paradigm in stroke patients, at least 22 patients per group were needed.

Data are presented as the mean±SD, except when stated otherwise. The measurement data were checked for normality (Shapiro–Wilk test). Chi-square (Levene's test) and one-way ANOVA were used to test for significance for normally distributed data; if the data were not normally distributed, the Kruskal–Wallis rank sum test was used. For count data, the chi-square test was used to test for significance. Two-way ANOVA was conducted to compare the main and interaction effects of HbO levels among groups [BFR, LL, HL] and ROIs [LDLPFC, RDLPFC, LM1, RM1, LPMC-SMA, RPMC-SMA]. If significant, least significant difference (LSD) post hoc tests were performed. All the data were analyzed via SPSS (IBM, V26), and $P < 0.05$ was considered to indicate statistical significance. A Pearson correlation analysis heatmap generated in R (version 4.0.2, Corrplot package) was used to assess the correlations between ΔHbO and changes in muscle performance.

## Results

### Participant characteristics

Among the 66 participants enrolled, 60 completed the 4-week intervention, with a 10% loss-to-follow-up rate. Six patients were excluded from further analysis because of significant fNIRS motion artifacts (n = 1), reports of unbearable muscle soreness or weakness (n = 2, HL), or dissociation during follow-up (n = 3). Missing data due to dropouts

were not imputed. The baseline characteristics were similar across the groups, with no statistically significant differences ($P > 0.05$; Table 1).

## Cortical activation

Two-way ANOVA revealed significant main effects of group ($F(2,57)=4.94$, $P=0.008$) and ROIs ($F(5,114)=13.51$, $P<0.001$) on $HbO_1$ levels. Moreover, there was a significant interaction effect between group and ROIs ($F(10,342)=2.83$, $P=0.042$). Multiple comparisons revealed that $HbO_1$ was significantly greater in the BFR and HL groups than in the LL group ($P=0.003$, $0.020$) but was not different between the BFR and HL groups ($P=0.511$). Compared with the M1 and PMC-SMA, the DLPFC had greater $HbO_1$ values in the bilateral hemispheres (all $P<0.001$), and no difference was noted between M1 and the PMC-SMA ($P=0.759$, $0.640$). Strong evidence indicated that $HbO_1$ levels were lower in RM1 than in LM1 ($P=0.008$) and in the RPMC-SMA than in the LPMC-SMA ($P=0.060$). The activation maps and HbO folding plots are shown in Figs 3a and 4a, and more detailed statistical results can be found in Table 2 and Supplementary S1 Table.

Following the 4-week intervention, no statistically significant difference in $HbO_2$ levels in the AH was noted compared with those in the unaffected hemisphere (UH) across the groups ($P>0.05$). The resulting brain activation maps are shown in Fig 3b. Two-way ANOVA for ΔHbO revealed significant main effects of intergroups ($F(2,57)= 3.88$, $P=0.021$) and ROIs ($F(5,114)=2.34$, $P=0.041$), and there was also a significant interaction effect between group and ROIs ($F(10,342)=7.50$, $P=0.001$). There was greater ΔHbO in both the BFR ($P=0.010$) and HL ($P=0.028$) groups than in the LL group; however, the ΔHbO did not differ between the BFR and HL groups ($P=0.711$). Comparisons between the AH and UH revealed that ΔHbO was significantly greater in the RM1 than in the LM1 ($P=0.034$), and that in the RPMC-SMA was greater than that in the LPMC-SMA ($P=0.031$), with no significant difference between the DLPFC on either side ($P=0.379$). The ΔHbO folding plots is shown in Fig 4b, and more detailed statistical results can be found in Table 2 and Supplementary S1 Table.

## Associations between cortical activation and muscle performance

The CSA, PT and FMLE scores did not differ between the groups before the intervention (all $P>0.05$). After four weeks, the CSA increased by 14.4%, 3.8%, and 18.3%, the PT increased by 70.6%, 8.1%, and 82.1%, and the FMLE scores increased by 40.2%, 14.2%, and 41.0% in the BFR, LL and HL groups, respectively. PT and FMLE scores significantly

**Table 1. Baseline anthropometric and physiological characteristics across groups.**

| Characteristics | BFR (n = 20) | LL (n = 20) | HL(n = 20) |
|---|---|---|---|
| Gender, male, n (%) | 12(60.0) | 11(55.5) | 12(60.0) |
| Age (years) | 61.7±9.6 | 60.9±10.3 | 61.0±8.0 |
| Educational level, less than ten years, n (%) | 13(65.0) | 12(60.0) | 10(50) |
| Affected Hemiplegia, left, n (%) | 15(75.0) | 12(60.0) | 13(65.0) |
| BMI | 24.3±4.8 | 25.0±3.3 | 26.4±3.1 |
| MMSE | 23.8±4.2 | 24.2±4.0 | 25.7±4.0 |
| Time poststroke* | 23.0(27.0) | 29.0(46.0) | 27.5(54) |
| Medical History | | | |
| Hypertension, n (%) | 12(57.1) | 14(66.7) | 11(73.3) |
| Diabetes, n (%) | 7(33.3) | 6(28.6) | 5(33.3) |
| Heart disease, n (%) | 4(19.0) | 4(19.0) | 3(20.0) |
| 1RM | 6.1±2.6 | 6.5±2.7 | 6.9±2.2 |

*Data are shown as median(interquartile range). BFR, blood flow restriction; LL, low load; HL, high load; BMI, body mass index; MMSE, Minimum Mental State Examination; 1RM, one-repetition maximum.

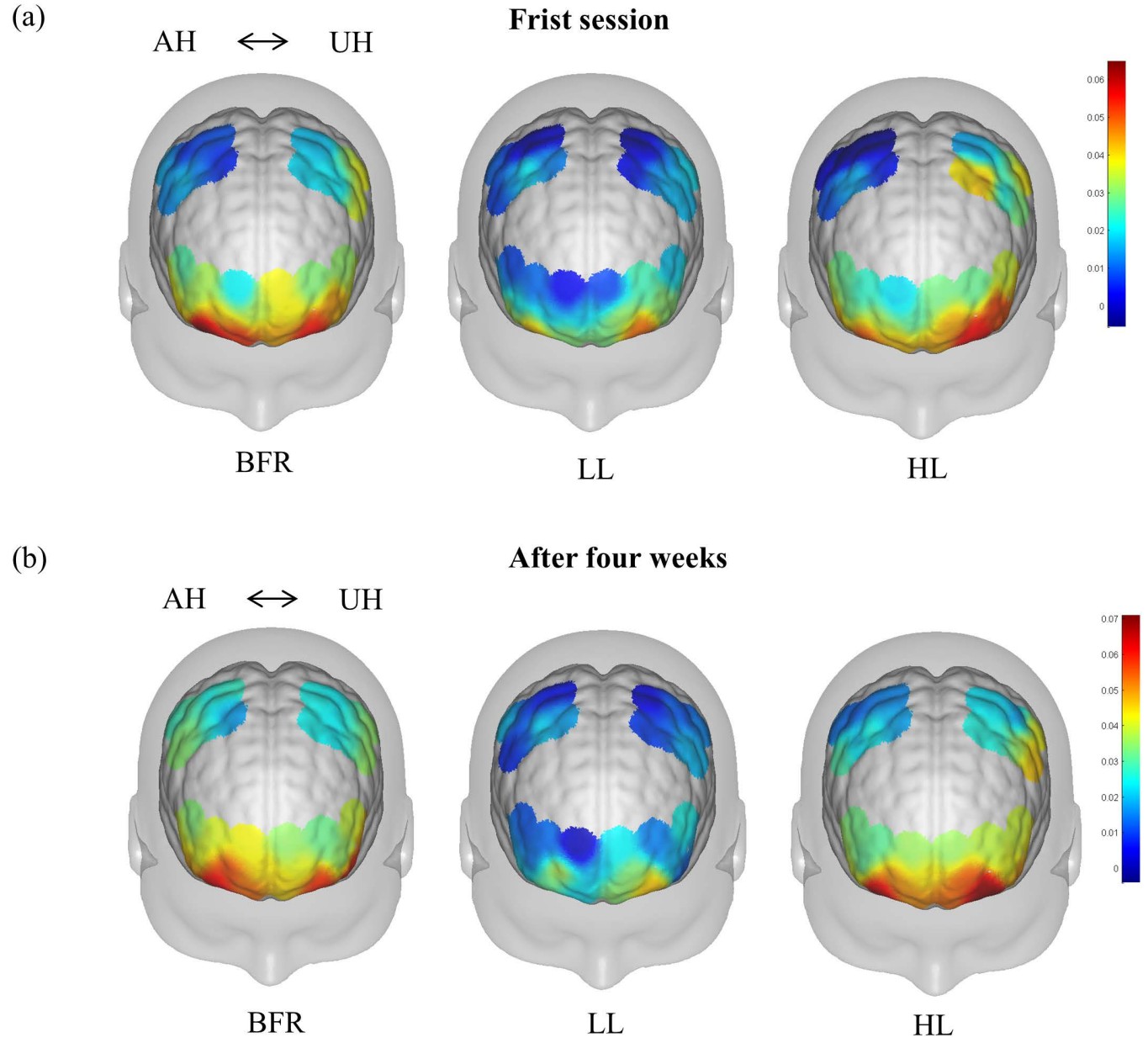

**Fig 3. Activation maps of HbO during (a) first session and (b) after four-week intervention.** AH, Affected hemisphere; UH, Unaffected hemisphere; BFR, blood flow restriction; LL,low load; HL, high load.

improved post-intervention in BFR and HL groups (all $P < 0.05$ vs. pre-intervention), with CSA showed no significant in either the BFR group ($P = 0.296$, pre- vs. post-intervention) or the HL group ($P = 0.144$). The LL group showed no significant changes in PT, FMLE, or CSA (all $P > 0.05$ pre- vs. post-intervention). Compared to the LL group, the HL group demonstrated significantly greater improvements in PT ($P < 0.001$) and FMLE ($P = 0.023$), while the BFR group showed enhanced PT only ($P < 0.001$ vs. LL). No significant differences in CSA were observed between HL, BFR, and LL groups ($P > 0.05$). Furthermore, direct comparisons between the BFR and HL groups revealed no statistically significant

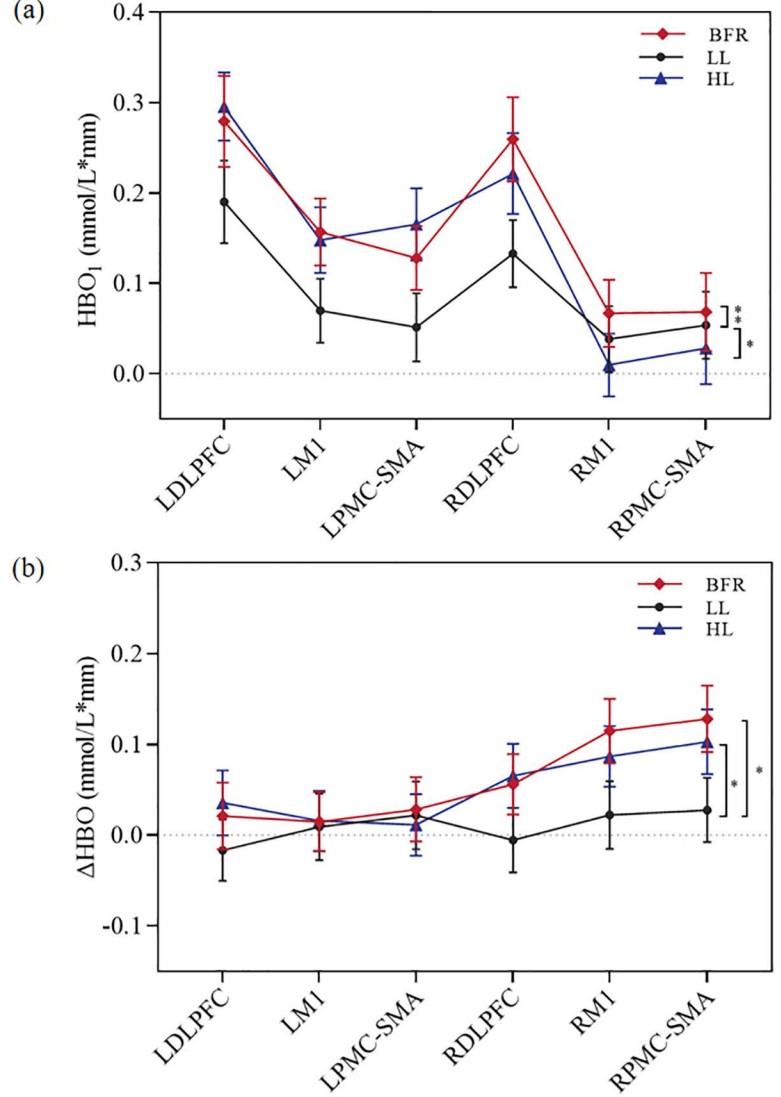

**Fig 4. Line graph of the difference of (a) HbO₁ and (b) Δ HbO between groups.** HbO₁ oxyhemoglobin concentration of first session; ΔHbO, pre-post change in oxyhaemoglobin concentration; BFR, blood flow restriction; LL,low load; HL, high load; L/RDLPFC, left/right dorsolateral prefrontal cortex; L/RM1, left/right primary motor cortex; L/RPMC-SMA, left/right premotor and supplementary motor cortex; **P<0.01, *P<0.05.

differences in CSA, PT, or FMLE scores (all *P*>0.05). Complete statistical results are presented in Table 3. A correlation headman (Fig 5) revealed that the ΔHbO values of RM1 and RPMC-SMA were positively correlated with changes in PT (mean *r*: 0.51 [range, 0.46–0.55]; all *P*<0.05) and FMLE scores (mean *r*: 0.54 [range, 0.48–0.62]; all *P*<0.05) in all groups. The correlations were found between the change in CSA and the ΔHbO of LM1, RDLPFC, RM1 and RPMC-SMA in HL group (mean *r*: 0.55 [range, 0.53–0.57]; all *P*<0.05).

## Discussion

To our knowledge, this is the first study to investigate the acute and chronic effects of LL, BFRT, and HLRT in the cerebral cortex and peripheral muscles via fNIRS in stroke patients. The main conclusions were as follows: (1) BFRT improved

**Table 2. Two-way ANOVA for HbO (mmol/L*mm) during the first session and pre-post change of four-week intervention.**

| ROIs | $HbO_1$ | | | ΔHbO | | |
|---|---|---|---|---|---|---|
| | BFR | LL | HL | BFR | LL | HL |
| **Unaffected** | | | | | | |
| LDLPFC | 0.28±0.23 | 0.19±0.21 | 0.30±0.17 | 0.02±0.16 | −0.02±0.15 | 0.04±0.16 |
| LM1 | 0.16±0.17 | 0.07±0.16 | 0.15±0.16 | 0.01±0.15 | 0.01±0.16 | 0.02±0.15 |
| LPMC-SMA | 0.13±0.16 | 0.05±0.17 | 0.17±0.18 | 0.03±0.16 | 0.02±0.17 | 0.01±0.15 |
| **Affected** | | | | | | |
| RDLPFC | 0.26±0.21 | 0.13±0.17 | 0.22±0.20 | 0.06±0.15 | −0.01±0.16 | 0.07±0.16 |
| RM1 | 0.07±0.17 | 0.04±0.16 | 0.01±0.16 | 0.11±0.16 | 0.02±0.17 | 0.09±0.15 |
| RPMC-SMA | 0.07±0.19 | 0.06±0.17 | 0.03±0.18 | 0.13±0.16 | 0.02±0.16 | 0.10±0.16 |
| **Intergroup effect ($F, P$)** | 4.94, 0.008 | | | 3.88, 0.021 | | |
| **ROIs effect ($F, P$)** | 13.51, <.001 | | | 2.34, 0.041 | | |
| **Interactive effect ($F, P$)** | 2.83, 0.042 | | | 7.50, 0.001 | | |

ROIs, region of interest; $HbO_1$, oxyhemoglobin concentration of first session; ΔHbO, pre-post change in oxyhaemoglobin concentration; BFR, blood flow restriction; LL,low load; HL, high load; L/RDLPFC, left/right dorsolateral prefrontal cortex; L/RM1, left/right primary motor cortex; L/RPMC-SMA, left/right premotor cortex and supplementary motor area;

**Table 3. Comparison of muscle performance for groups before and after four-week intervention.**

| Variables | BFR | | LL | | HL | |
|---|---|---|---|---|---|---|
| | before | after | before | after | before | after |
| CSA | 1.88±0.76 | 2.15±0.85 | 1.84±0.71 | 1.91±0.74 | 1.97±0.69 | 2.33±0.83 |
| PT | 17.88±6.96 | 30.51±9.47*# | 17.92±7.50 | 19.37±8.09 | 18.61±6.63 | 33.89±8.20*# |
| FMLE | 18.05±8.58 | 25.30±7.24* | 19.00±7.53 | 21.70±7.58 | 19.25±7.43 | 27.15±6.95*# |

CSA, cross-sectional area; PT, peak torque; FMLE, Fugl-meyer motor assessment of lower extremities. BFR, blood flow restriction; LL,low load; HL, high load; *$P<0.05$, for within-group pre- to post-intervention differences; #$P<0.05$,for for between-group comparisons vs. the LL group.

acute and chronic motor-related cortical activation of AH, muscle strength and performance more than LL alone did in stroke survivors; (2) cerebral cortical activation and muscle performance for BFRT and HLRT were similar when lower-limb muscle groups with intermittent BFRT were used; and (3) the M1 and PMC-SMA may be more crucial for active modulation during exercise following a stroke than the DLPFC is. Alterations in the M1 and PMC-SMA activities of the AH and muscle performance were strongly associated.

## Acute neurological adaptation

The acute effects of exercise on central excitability depend on the equilibrium between the oxygen supply and consumption; either a reduction in the oxygen supply or an increase in oxygen consumption may result in a decrease in HbO [23]. After brain infarction, impaired cerebrovascular reactivity results in reduced blood oxygen level-dependent signaling during exercise in the AH. Our results support previous studies of acute increases in brain oxygenation during HL exercise in healthy participants by demonstrating that stroke patients exhibit a further increase in cortical HbO with a significant increase in exercise load (up to 80% 1RM) [24,25]. Muscle oxygenation and brain oxygenation respond differently to exercise, according to a recent systematic review of 20 studies in healthy subjects [26]. Muscle $O^2$ utilization increases (indicated by [HHb increase]), whereas brain $O^2$ delivery increases (shown by [HbO increase]) with increasing exercise load. Importantly, brain oxygenation plateaus or decreases with fatigue [26]. Such acute high exercise intensities are associated with hyperventilation-induced reductions in arterial $CO^2$ tension, resulting in cerebral vasoconstriction and

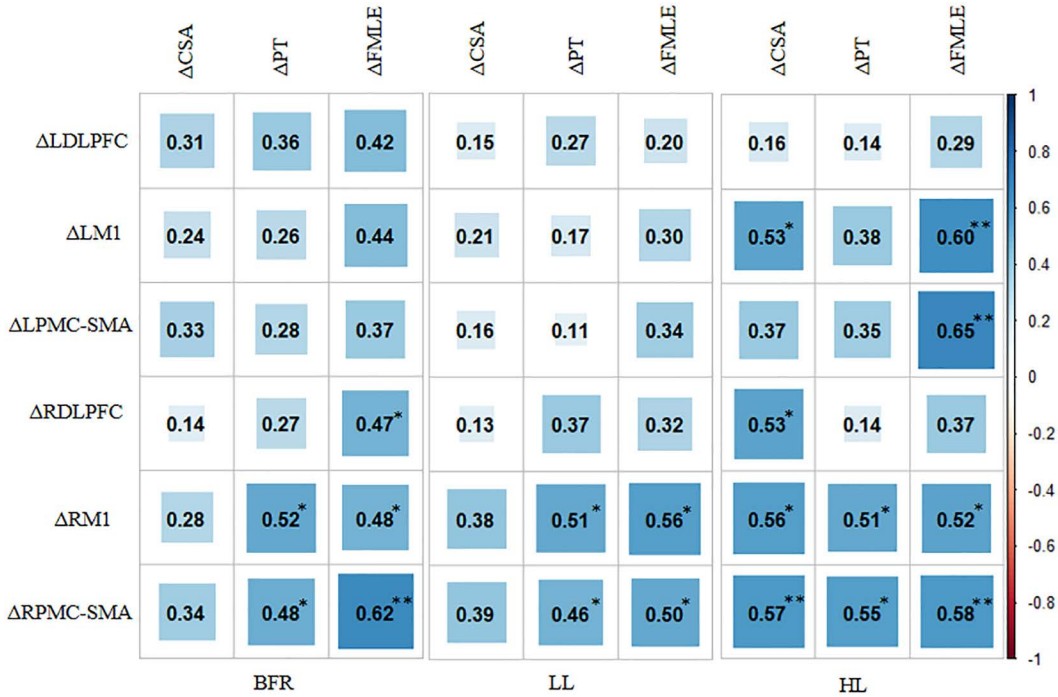

**Fig 5. Correlation heatmap of ΔHbO in cortical regions of interest with muscle gains.** ΔHbO, pre-post change in oxyhaemoglobin concentration; CSA, cross-sectional area; PT, peak torque; FMLE, Fugl-Meyer lower extremity; BFR, blood flow restriction; LL,low load; HL, high load; L/RDLPFC, left/right dorsolateral prefrontal cortex; L/RM1, left/right primary motor cortex; L/RPMC-SMA, left/right premotor cortex and supplementary motor area; **$P < 0.01$, *$P < 0.05$.

diminished cerebral blood flow [27]. Thus, increased HbO levels during HLRT can be interpreted as enhanced central activity, whereas the transient paradigm of the present study may not have induced an inhibitory effect on fatigue.

Limited and inconsistent data exist on the cortical response to BFRT in stroke patients. There is evidence that in healthy individuals, corticospinal excitability is greater in the short term (1 min after BFRT [12]) or long term (up to 60 min after BFRT [13]) following LL-BFR than in LL when the biceps brachii [13] and tibialis anterior (TA) [12] are subjected to different exercise regimens. Those findings support the results of the present study that the addition of BFR to LL-RT significantly altered the acute response of the brain. In contrast, in the other two studies, neuromuscular activity was examined before, immediately after, 10 min after and 20 min after TA dorsiflexion exercise in stroke patients and healthy participants using BFR, and no significant differences were found in the amplitude of motor-evoked potentials recorded by TMS, short-interval intracortical inhibition, intracortical facilitation or peripheral sEMG amplitudes compared with the exercise-only group [28,29]. Notably, most of the included studies adopted a pre-post design, which, unfortunately, may have reduced the ability to demonstrate neuroplasticity during the training period. Notably, the use of TMS for measuring neuronal excitability has inherent limitations.

## Chronic neurological adaptation

Consistent with prior findings, we discovered that BFRT or HLRT greatly enhanced muscular function in stroke patients over time. Along with functional improvements, we found a tendency toward increased AH motor cortex activity in both the BFR and HL groups compared with the LL group. This finding represents the requirement for the patient's CNS to increase activation intensity over time, for example, by recruiting more motor neurons and increasing neural signaling frequency,

to fulfill the external demand for increased force production. With respect to the chronic effects of BFRT, Cook et al. [30] reported that following six weeks of leg press and knee extension training in healthy young adults, neural responses (central activation, twitch torque parameters, post-activation augmentation, and half-relaxation time) were similar in the HLRT (70%1RM) and BFRT (20%1RM) groups. Sprick et al. [31] reported that cyclic BFRT (20%1RM) and HLRT (65%1RM) produce similar cerebrovascular responses and reduced sympathetic responses during leg press exercise in healthy individuals. Two additional chronic BFRT studies revealed that 6 weeks [32] and 8 weeks [33] of BFRT significantly increased the sEMG amplitude of the lateral femoral muscle. These comparable central-peripheral effects align with our study results, implying that the chronic effects of BFRT on peripheral muscle function may not be exclusively dependent on rapid increases in muscle hypertrophy but rather on central adaptation.

BFRT is also thought to regulate blood lactate and brain-derived neurotrophic factor (BNDF) levels and decrease muscle growth inhibitor expression in a manner similar to HLRT [34]. Charalambous et al. [35] reported rapidly increasing blood lactate levels during HL seated ergometer training in chronic stroke patients. Oliveira et al. [36] also reported that BFRT increased blood lactate accumulation by 16% and maximum power output by 15%, and these values did not differ significantly from those of the HLRT groups. Lactate is essential for a variety of brain functions, including neuronal metabolism, neuroprotection, and long-term memory formation [37]. The increase in brain lactate after exercise corresponds with changes in peripheral blood lactate [38], and it is correlated with a lower motor threshold or an increase in motor cortical excitability [39]. Studies of intravenous lactate infusion in healthy adults have shown that increased lactate leads to elevated serum BDNF [39,40], which is thought to bridge the gap between exercise and neuroplasticity [41], increase corticospinal excitability and decrease intracortical inhibition [42]. Furthermore, systematic reviews have demonstrated that the cardiorespiratory fitness hypothesis, which posits that cardiorespiratory fitness enhances cerebrovascular structure and function, can be used to explain the relationship between higher levels of physical activity and/or cardiorespiratory fitness and higher levels of cortical activity in cross-sectional studies and long-term exercise studies [43].

## Region-specific changes in brain oxygenation

Cortical activation by BFRT and HLRT was more pronounced in the motor cortex-related areas. This finding demonstrates that the compensatory rearrangement of the cerebral cortex, both structurally and functionally, gradually narrows down to the motor areas of the AH. Moreover, the newly formed neural functional network gradually reestablishes and converges to the innervation pattern of the normal condition. Rodent ischemic stroke trials have shown that poststroke vascular remodeling facilitates behavioral recovery after stroke by restoring blood flow to the peri-infarct cortex [2]. The greater ΔHbO in M1 than in other regions confirm the dominant role of this region in stroke recovery, which is consistent with M1's dominant role in transmitting motor commands to the corticospinal tract, as previously reported by Theresa et al. [44]. Regarding the PMC-SMA, previous research has demonstrated the critical role of the PMC in complex motor control (involving interjoint coordination and visuomotor integration) in stroke patients [44]. The pedaling used in the present study was undoubtedly a complex sequential movement involving multiple joints; hence, the activation of this region, similar to that of M1, reaffirms its importance in locomotion. Moreover, we hypothesized that BFRT involves some form of motor area modulation, similar to HLRT. It has been proposed that high levels of external compression, reduced blood flow, and an ischemic/hypoxic intramuscular environment all potentially play a role in stimulating an increase in muscle activation via group III and IV muscle afferents. Sensory feedback to cortical and/or subcortical areas during exercise and under ischemic/hypoxic conditions has been proposed to alter muscle activation and corticomotor excitability [13]. The evidence from the present study further supports a potential role of group III and IV muscle afferents in modulating corticomotor excitability with BFRT.

## Association between central and peripheral performance

For peripheral performance, both HLRT and BFRT produced similar improvements in lower extremity strength and motor performance. Several studies focused on peripheral muscle excitability have reported increased peripheral sEMG during

BFRT [29,32,45,46], but the increased sEMG amplitude obtained with BFR is usually not greater than that observed with HL [32,47]. Previous fMRI and fNIRS studies of upper limb movements (e.g., pinching, grasping, lifting the right hand) have also shown that motor cortex activation is positively correlated with force output [24]. Thus, we hypothesized that the similar central versus peripheral gains in HLRT and BFRT may involve different activation patterns, with the latter being late onset, which reflected by the relatively muted growth in CSA, PT, and FMLE in BFR group. Indeed, Fatela et al. [45] demonstrated that HLRT results in maximal muscle activation in the early stages of the training process, whereas muscle activation increases gradually with BFRT. Another study also showed that compared with non-BFRT at 20%1 RM or 50%1 RM, BFRT at 20%1 RM resulted in a significant reduction in quadriceps blood flow during exercise, with BFRT outperforming the other two regimens, explaining the post-exercise anabolic potential of BFRT [48]. Notably, the RF-CSA benefit in the affected lower extremity of stroke individuals after short-term BFRT was not significant compared with that of muscle strength and performance. This may be explained by the fact that during the first 2–4 weeks of RT, most of the gains in strength are primarily related to neural adaptations [49]. These neural adaptations were evidenced by a positive correlation between peripheral muscle gains and motor cortical activation in the AH in our study. This finding may be supported by the fact that Simranjit et al. [50] reported the inhibition of EMG activity in many of the major lower-limb muscles during cycling by inhibitory TMS acting on the motor cortex, implying a direct contribution of the motor cortex to lower-limb muscle function.

## Limitations

The limitations of this work should be acknowledged. First, regarding research methodology, fNIRS data interpretation requires caution given its indirect measurement nature; residual noise may influence findings despite rigorous preprocessing. Second, variability in BFRT protocols (continuous/intermittent application, cuff specifications, pressure levels, duration) and exercise parameters (muscle groups, loading intensity) limits cross-study comparability. Current arterial occlusion pressure (AOP) recommendations span 40–80% [51], while applied pressures range widely (50–230 mmHg across literature). Our use of standardized 200 mmHg pressure – implemented for safety and operational feasibility in stroke patients, which may impact generalizability. Third, our analytical approach followed per-protocol rather than intention-to-treat (ITT) principles. Future investigations should incorporate ITT analysis to evaluate robustness against attrition bias and maintain randomization integrity. Fourth, the cohort focused exclusively on mild-to-moderate stroke patients (muscle strength ≥grade 3), whereas individuals with severe hemiplegia (<grade 3) commonly encountered in clinical practice were excluded, limiting applicability to more impaired populations.

## Conclusions

The present study highlights the superiority of BFRT and HLRT over LLRT alone for the development of central activation and muscle performance through neural adaptations rather than hypertrophic changes. Training programs should therefore prioritize neuromodulatory stimuli via targeted exercise variable manipulation. For coaches and therapists, implementing BFRT (30% 1RM with occlusion) or HLRT (80% 1RM) during early rehabilitation can effectively enhance neural drive, while real-time fNIRS monitoring allows personalized adjustment of exercise intensity. For patients, supervised BFRT/HLRT combined with functional tasks promotes synergistic neural-muscular improvements, with progress tracked through both wearable fNIRS-based brain activation and strength metrics. For researchers, optimizing BFRT parameters to maximize long-term neuroplasticity and potential hypertrophy effects, alongside validating HbO as a rehabilitation efficacy biomarker, will advance precision rehabilitation strategies. Long-term and chronic adaptations are particularly relevant for clinical rehabilitation, especially for stroke patients, who aim to achieve greater gains on the affected side.

## Supporting information

**S1 Appendix. CONSORT Checklist.**
(PDF)

**S2 Appendix. Protocol.**
(PDF)

**S1 Table. Results of multiple comparisons (LSD post-hoc tests) for group and region of interest effects.**
(DOCX)

## Acknowledgments

This work was conducted at the Neurological Rehabilitation Centre of the First People's Hospital of Lianyungang. We thank all the patients and their families for their cooperation and the therapists for their work. We thank Yang Yunji, Ding Zhihao and Gao Man for data acquisition support, and Liu Siyin for acquisition paradigm guidance and image interpretation. We thank Danyang Huichuang Brain Science Research Centre for providing instrument training and academic support.

## Author contributions

**Conceptualization:** Xinyu Bai, Yunyun Zhang.

**Data curation:** Xinyu Bai, Yunyun Zhang.

**Formal analysis:** Xinyu Bai, Yunyun Zhang.

**Investigation:** Yang Xie, Weiwei Cao.

**Methodology:** Xinyu Bai.

**Resources:** Weiwei Cao.

**Software:** Yunyun Zhang.

**Validation:** Xinyu Bai.

**Writing – original draft:** Xinyu Bai.

**Writing – review & editing:** Yunyun Zhang, Yonggang Zhu.

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
