## [Decision Letter · Decision Letter 0]

11 Apr 2025

Thank you for submitting your manuscript to PLOS ONE. After careful consideration, we feel that it has merit but does not fully meet PLOS ONE’s publication criteria as it currently stands. Therefore, we invite you to submit a revised version of the manuscript that addresses the points raised during the review process.

We look forward to receiving your revised manuscript.

Kind regards,

Miray Budak

Academic Editor

PLOS ONE

Journal Requirements:

2. If any supporting files for review show as item type ‘other’ please change to item type ‘supporting info’ as the reviewer does not have access to these ’other’ files.

5. Please amend either the title on the online submission form (via Edit Submission) or the title in the manuscript so that they are identical.

Reviewers' comments:

Reviewer's Responses to Questions

**Comments to the Author**

1. Is the manuscript technically sound, and do the data support the conclusions?

Reviewer #1: Partly

Reviewer #2: Yes

2. Has the statistical analysis been performed appropriately and rigorously?

Reviewer #1: No

Reviewer #2: Yes

3. Have the authors made all data underlying the findings in their manuscript fully available?

Reviewer #1: Yes

Reviewer #2: Yes

4. Is the manuscript presented in an intelligible fashion and written in standard English?

Reviewer #1: Yes

Reviewer #2: Yes

Reviewer #1: The manuscript is generally well presented statistically. However, there is some detail missing as noted below.

The sample size calculation with power considerations as well as the statistical analysis plan is consistent from the supplemental protocol to the written manuscript. The analysis elements are routine and applied appropriately for a three group design. They include the Shapiro-Wilk test and Levene's test assess the normality and homogeneity of variance of the data. For data that follow a normal distribution, one-way ANOVA was used to test differences; otherwise, they say that the Kruskal-Wallis rank-sum test will be used. All of the results reporting appears to be normality based. Was there any non parametric reporting or need for such?

Chi-square tests were used for categorical data. Two-way ANOVA was used to evaluate the main and interaction effects of group and region of interest on HbO mean. If significant, LSD post-hoc tests will be conducted. For any post hoc comparisons with p-values, what was the post hoc test used?

Pearson correlation analysis was used to assess the correlation between cortical HbO and motor performance. This reporting was reasonable.

The sample size called for 22 per group and 15% dropout rate. Table 1 shows 20 per group. Did all 20 per group complete the analysis in Tables 2 and 3? If not, is there a missing data issue?

Why is there no p-value for group comparisons (pre-post) in Table 3? It is not entirely clear in the paragraph discussing the results of Table 3.

The investigators have clearly listed the limitations in the discussion. Given the detection time points schematic in the protocol, why is there not a time factor for data (re-measurement) in the overall ANOVA?

Reviewer #2: Dear Authors,

Thank you for your efforts to shape this carefully crafted study. In order to increase the quality of the manuscript, you can find my recommendations below:

The objective sentence of the manuscript must be strengthened and must give more context about the importance and possible beneficial effects of the study into the related literature.

At the end of the conclusions section, please provide point to point recommendations coaches, patients and sport scientists about the potential applications of your results to improve the quality of exercise sessions in ischemic stroke patients.

Best regards,

**Do you want your identity to be public for this peer review?** For information about this choice, including consent withdrawal, please see our Privacy Policy

Reviewer #1: No

Reviewer #2: No

---

## [Author Response · Author response to Decision Letter 1]

12 May 2025

Dear Editors

We gratefully appreciate the editors and all reviewers for their time spend making positive and constructive comments. Those comments are all valuable and helpful for revising and improving our article entitled “Cerebral cortical activation and muscle performance during blood flow restriction training after ischemic stroke: a randomised functional near-infrared spectroscopy study” (ID: PONE-D-25-08906), as well as the important guiding significance to our research.

We have studied the comments carefully and have made corrections which we hope meet with approval. Point-to-point responses to academic editor and reviewers are listed below. Revised portions are highlighted in the manuscript.

Thank you and best regards.

Responses to academic editor (Responses by the author are in blue)

Academic editor Comments:

1. When submitting your revision, we need you to address these additional requirements. Please ensure that your manuscript meets PLOS ONE's style requirements, including those for file naming. If any supporting files for review show as item type‘other’please change to item type‘supporting info’as the reviewer does not have access to these ’other’ files.

Response: Thank you for your feedback and the opportunity to revise our manuscript. We have carefully addressed the additional requirements outlined in your message. Specifically, we confirm the following:

1. PLOS ONE Style Requirements:

The manuscript has been reformatted to fully comply with PLOS ONE’s style guidelines, including references, figure/table labeling, and ethical/legal standards.

All supporting files (figures, tables, supplemental materials) have been checked for formatting consistency.

2. File Naming Conventions:

All submission files (manuscript, figures, supplementary materials) have been renamed according to PLOS ONE’s requirements.

3. Supporting File Item Types:

Any supporting files previously categorized as “other” in the submission system have been updated to the correct item type “supporting information” to ensure reviewers have full access.

We have double-checked the revised submission to ensure all editorial and stylistic criteria are met. Should further adjustments or clarifications be needed, please don’t hesitate to contact us.

Response: Thank you for your kind remind. We confirm that our submission includes the minimal data set required to replicate all study findings. All raw experimental data, associated metadata, and methodological details are provided within the Supporting Information files.

Response: We confirm that the corresponding author’s ORCID iD has been validated in Editorial Manager following the journal’s requirements. The ORCID record is now fully linked and accessible in the system. Thank you for the guidance.

Response: Thank you for your note. We confirm that the manuscript title has been revised to match exactly the title in the online submission system. Minor edits were made during CONSORT 2010 checklist alignment (e.g., clarifying study design), and both versions are now consistent. We apologize for the oversight and appreciate your guidance.

Responses to Reviewers (Responses by the author are in blue)

Reviewer #1:

1. Comments: The sample size calculation with power considerations as well as the statistical analysis plan is consistent from the supplemental protocol to the written manuscript. The analysis elements are routine and applied appropriately for a three group design. They include the Shapiro-Wilk test and Levene's test assess the normality and homogeneity of variance of the data. For data that follow a normal distribution, one-way ANOVA was used to test differences; otherwise, they say that the Kruskal-Wallis rank-sum test will be used. All of the results reporting appears to be normality based. Was there any non parametric reporting or need for such?

Response: Thank you so much for your careful review and thoughtful feedback! We sincerely appreciate your attention to detail. Regarding your question, we did use non-parametric methods where assumptions were not met. For example, Time poststroke (Table 1) did not follow a normal distribution (Shapiro-Wilk test, p < 0.05), so we applied the Kruskal-Wallis test, as stated in our statistical plan. These results are reported in Section Result (Line181-182) of the manuscript. Your insights are invaluable, and we are happy to provide additional details if needed!

2. Chi-square tests were used for categorical data. Two-way ANOVA was used to evaluate the main and interaction effects of group and region of interest on HbO mean. If significant, LSD post-hoc tests will be conducted. For any post hoc comparisons with p-values, what was the post hoc test used?

Response: Thank you for your advice. For significant main or interaction effects identified in the two-way ANOVA (group × region of interest), LSD test was applied for pairwise multiple comparisons. The validity of this approach was ensured by confirming the homogeneity of variance assumption. Detailed statistical procedures for multiple comparisons are described in the Methods section (Lines 170-172); Key comparison results are reported in the Results section (Lines 189-193 and Lines 202-206); Complete numerical outcomes of all pairwise comparisons (including mean differences, p-values, and confidence intervals) are provided in Supplementary Table S3.

3. Pearson correlation analysis was used to assess the correlation between cortical HbO and motor performance. This reporting was reasonable.

Response: Thank you sincerely for your positive feedback and careful evaluation of our analysis!

4. The sample size called for 22 per group and 15% dropout rate. Table 1 shows 20 per group. Did all 20 per group complete the analysis in Tables 2 and 3? If not, is there a missing data issue?

Response: Thank you for raising this critical point. We clarify as follows:

1. Sample size validity: The final analyzed sample (n=20 per group, total n=60) exceeds the pre-specified minimum requirement of 57 subjects stated in the Statistical Power Calculation subsection (Methods, Section Statistical Analysis).

2. Dropout handling transparency: As noted in the revised Results section: "Missing data due to dropouts were not imputed." Attrition details (e.g., discharge, muscle soreness) are fully documented in Fig 1 (CONSORT flowchart) and Table 1 (baseline data of analyzed cohorts)

3. Analysis consistency: All results in Tables 2-3 strictly reflect complete cases (n=20 per group). No intention-to-treat (ITT) analysis was performed given the lack of post-dropout endpoint data, aligning with our pre-registered complete-case protocol.

We appreciate the opportunity to strengthen methodological clarity.

5. Why is there no p-value for group comparisons (pre-post) in Table 3? It is not entirely clear in the paragraph discussing the results of Table 3.

Response: We sincerely appreciate your careful review of the statistical reporting. We have made the following clarifications and revisions to address this concern.

The asterisk (*) in Table 3 denotes significant within-group pre-post differences (P<0.05), while the hash symbol (#) indicates significant between-group differences (P<0.05) after intervention. These definitions are now explicitly stated in the table footnote (see revised Table 3 caption).

In the revised Results section (Lines 226-234), we have added detailed statistical descriptors: “PT and FMLE scores significantly improved post-intervention in BFR and HL groups (all P<0.05 vs. pre-intervention), with CSA showed no significant in either the BFR group (P=0.296, pre- vs. post-intervention) or the HL group (P=0.144). The LL group showed no significant changes in PT, FMLE, or CSA (all P>0.05 pre- vs. post-intervention). Compared to the LL group, the HL group demonstrated significantly greater improvements in PT (P<0.001) and FMLE (P=0.023), while the BFR group showed enhanced PT only (P<0.001 vs. LL). No significant differences in CSA were observed between HL, BFR, and LL groups (P> 0.05). Furthermore, direct comparisons between the BFR and HL groups revealed no statistically significant differences in CSA, PT, or FMLE scores (all P>0.05). Complete statistical results are presented in Table 3.

6. The investigators have clearly listed the limitations in the discussion. Given the detection time points schematic in the protocol, why is there not a time factor for data (re-measurement) in the overall ANOVA?

Response: Thank you for raising this critical methodological point. Below are our detailed responses regarding the temporal factor analysis:

1. Rationale for not applying repeated-measures ANOVA

Hemodynamic data were collected only at two time points (baseline: first training session; endpoint: last training session). As repeated-measures ANOVA generally requires ≥3 time points to adequately model temporal trends and within-subject variability, we adopted the following alternative strategies to assess time effects while avoiding overinterpretation of limited longitudinal data:

2. Alternative analytical approaches for temporal effects

(1) Baseline brain activation: Initial neural responses were analyzed using two-way ANOVA (Group × ROI) at the first training session;

(2) Delta-value analysis: We computed per-subject ΔHbO (last session minus first session) and examined group/ROI effects on these delta values via the same ANOVA framework;

(3) Clinical correlation validation: Pearson/Spearman correlations between ΔHbO and concurrent changes in muscle mass/strength/performance were performed (Fig 5), establishing temporal coherence between neuroadaptation and functional outcomes.

We fully agree that the two-timepoint design limits granular temporal modeling. Future studies will incorporate intermediate assessments to enable more sophisticated longitudinal analyses.

Reviewer #2 Comments

1. The objective sentence of the manuscript must be strengthened and must give more context about the importance and possible beneficial effects of the study into the related literature.

Response: Thank you for emphasizing the need to contextualize our study's objectives within the broader literature. We have substantially strengthened the objective statement and background rationale in the Introduction (Lines 25-55), with objective sentence revisions outlined below: To resolve the critical knowledge gaps in BFRT-induced neuroplasticity, we conducted a randomized controlled trial integrating real-time fNIRS with muscle strength-structure-function profiling. We aimed to: (a) Compare acute-phase cortical activation (prefrontal-motor HbO) across BFRT (30%1RM+occlusion), HLRT (80%1RM), and LLRT (30%1RM); (b) Investigate temporal associations between HbO changes and improvements in muscle strength, hypertrophy , and motor function; (c) Validate if BFRT induces HLRT-equivalent neuroplasticity with 50% lower mechanical loads, prioritizing central drive over peripheral adaptations. These revisions explicitly anchor our objectives within unresolved theoretical debates (e.g., central-peripheral interplay) and unmet clinical needs (e.g., load-intolerant patients). We appreciate your guidance in elevating the manuscript's scholarly impact.

2. At the end of the conclusions section, please provide point to point recommendations coaches, patients and sport scientists about the potential applications of your results to improve the quality of exercise sessions in ischemic stroke patients.

Response: Thank you for highlighting the importance of translational applications. We have revised the Conclusions section (Lines 379-386) to include targeted recommendations for stakeholders, as detailed below: For coaches and therapists, implementing BFRT (30% 1RM with occlusion) or HIRT (80% 1RM) during early rehabilitation can effectively enhance neural drive, while real-time fNIRS monitoring allows personalized adjustment of exercise intensity. For patients, supervised BFRT/HIRT combined with functional tasks promotes synergistic neural-muscular improvements, with progress tracked through both wearable fNIRS-based brain activation and strength metrics. For researchers, optimizing BFRT protocols to amplify long-term neuroplasticity and muscle hypertrophy, alongside validating HbO as a rehabilitation efficacy biomarker, will advance precision rehabilitation strategies.

We tried our best to improve the manuscript and made some changes in the manuscript. We think that our manuscript has greatly improved after revisions, and we would like to submit the revised manuscript for publication in Plos one. We appreciate the reviewers’ warm work earnestly and hope that the correction will meet with approval. Once again, thank you very much for your comments and suggestions.

Best wishes,

May 8th, 2025

---

## [Decision Letter · Decision Letter 1]

24 Jun 2025

Dear Dr. zhu,

Thank you for submitting your manuscript to PLOS ONE. After careful consideration, we feel that it has merit but does not fully meet PLOS ONE’s publication criteria as it currently stands. Therefore, we invite you to submit a revised version of the manuscript that addresses the points raised during the review process.

We look forward to receiving your revised manuscript.

Kind regards,

Miray Budak

Academic Editor

PLOS ONE

Journal Requirements:

Reviewers' comments:

Reviewer's Responses to Questions

**Comments to the Author**

Reviewer #1: All comments have been addressed

Reviewer #3: All comments have been addressed

Reviewer #4: All comments have been addressed

2. Is the manuscript technically sound, and do the data support the conclusions?

Reviewer #1: (No Response)

Reviewer #3: Yes

Reviewer #4: Yes

3. Has the statistical analysis been performed appropriately and rigorously?

Reviewer #1: (No Response)

Reviewer #3: Yes

Reviewer #4: Yes

4. Have the authors made all data underlying the findings in their manuscript fully available?

Reviewer #1: (No Response)

Reviewer #3: Yes

Reviewer #4: Yes

5. Is the manuscript presented in an intelligible fashion and written in standard English?

Reviewer #1: (No Response)

Reviewer #3: No

Reviewer #4: Yes

Reviewer #1: (No Response)

Reviewer #3: The authors present a well-designed, double-blind randomized controlled trial that innovatively combines blood-flow-restriction and high-load resistance training with real-time functional near-infrared spectroscopy (fNIRS) monitoring in post-stroke rehabilitation. The integration of cortical HbO dynamics with muscle performance outcomes is timely and clinically relevant. The manuscript answers an important translational question and is, in my view, publishable after the minor revisions detailed below.

1. The title entered in Editorial Manager is still in Chinese, whereas the English title appears on the manuscript title page. Update the submission-system title (currently in Chinese) to match the English manuscript title to avoid downstream indexing errors.

2. There are some typographic errors, data-availability formalities as well as the abbreviation and acronym definitions. Please try to find all, and fix them.

- Both HLRT and HIRT are used for high-load training, choose one acronym and apply consistently.)

- “HBO” appears several times; accepted symbol is HbO, uniformly replace with HbO.

- Run spell-check to catch “frist”.

- Add “(fNIRS)” in the Abstract, use the acronym after introduced at first for consistency.

- Standardise percentage formatting (use either 70 % or 70%, not both)

A professional language edit is strongly advised to eliminate residual slips and improve readability.

Reviewer #4: The introduction mostly references general stroke and resistance training literature but could benefit from deeper integration of BFRT-specific neuroimaging studies, especially in clinical populations.

Integrate 1–2 key studies that examined BFRT’s cortical effects (e.g., via TMS, EEG, fMRI) in healthy or stroke subjects to better frame the study’s novelty.

Using a fixed pressure (200 mmHg) for all participants may lead to uneven BFRT stimulus, especially considering individual differences in limb size and vascular resistance.

Clarify that short-term strength gains likely reflect neural adaptation rather than hypertrophy.

Include ITT as sensitivity analysis in future studies or at least discuss this limitation.

**Do you want your identity to be public for this peer review?** For information about this choice, including consent withdrawal, please see our Privacy Policy

Reviewer #1: No

Reviewer #3: No

Reviewer #4: No

---

## [Author Response · Author response to Decision Letter 2]

2 Jul 2025

Dear Editors

We gratefully appreciate the editors and all reviewers for their time spend making positive and constructive comments. Those comments are all valuable and helpful for revising and improving our article entitled “Cerebral cortical activation and muscle performance during blood flow restriction training after ischemic stroke: a randomised functional near-infrared spectroscopy study” (ID: PONE-D-25-08906R1), as well as the important guiding significance to our research.

We have studied the comments carefully and have made corrections which we hope meet with approval. Point-to-point responses to reviewers are listed below. Revised portions are highlighted in the manuscript.

Thank you and best regards.

Responses to Reviewers (Responses by the author are in blue)

Reviewer #3:

1. Comments: The title entered in Editorial Manager is still in Chinese, whereas the English title appears on the manuscript title page. Update the submission-system title (currently in Chinese) to match the English manuscript title to avoid downstream indexing errors.

Response: We sincerely apologize for this administrative oversight. The English manuscript title has now been updated in Editorial Manager to match the title page (current Chinese title removed). This correction ensures consistency across all submission records to prevent downstream indexing issues. Thank you for catching this critical detail.

2.  Comments: There are some typographic errors, data-availability formalities as well as the abbreviation and acronym definitions. Please try to find all, and fix them.

- Both HLRT and HIRT are used for high-load training, choose one acronym and apply consistently.)

- “HBO” appears several times; accepted symbol is HbO, uniformly replace with HbO.

- Run spell-check to catch “frist”.

- Add “(fNIRS)” in the Abstract, use the acronym after introduced at first for consistency.

- Standardise percentage formatting (use either 70 % or 70%, not both)

Response: We thank the reviewer for these critical technical corrections. Throughout the manuscript, we have: standardized the acronym to HLRT (removing all 'HIRT' instances); corrected 'HBO' to 'HbO' for oxyhemoglobin notation; fixed typographical errors (e.g., 'frist'→'first') via full spell-check; added '(fNIRS)' after the first abstract mention; normalized percentage formatting to 70%. These updates ensure terminological precision and formatting consistency across the submission.

Reviewer #4

1. Comments The introduction mostly references general stroke and resistance training literature but could benefit from deeper integration of BFRT-specific neuroimaging studies, especially in clinical populations.

Integrate 1–2 key studies that examined BFRT’s cortical effects (e.g., via TMS, EEG, fMRI) in healthy or stroke subjects to better frame the study’s novelty.

Response: We thank the reviewer for highlighting the need for deeper integration of BFRT-specific neuroimaging literature. In response, we have incorporated three pivotal studies examining BFRT's cortical effects in the Introduction (Lines 41-46): Crucially, BFRT demonstrates amplified cortical responses in healthy individuals, with TMS showing significantly greater corticospinal excitability during upper- or lower-limb BFRT versus low load (LL) RT[12, 13]. Specifically, functional near-infrared spectroscopy (fNIRS) reveals optimal motor cortex activation during squatting with moderate-pressure BFRT (250 mmHg), outperforming both non-restricted (0 mmHg) and alternative pressure conditions (150 and 300 mmHg)[7].

2. Comments Using a fixed pressure (200 mmHg) for all participants may lead to uneven BFRT stimulus, especially considering individual differences in limb size and vascular resistance.

Response: We appreciate the reviewer's valid concern regarding individualized pressure prescription. Our pressure selection rationale is now explicitly stated in the Methods section (Lines 96-99): Previous studies in healthy subjects have indicated that a pressure of 250 mmHg is optimal for achieving the greatest degree of cerebral activation[7]. Considering that circulation may be impaired in stroke patients, we set the pressure value at 80% (200 mmHg). Furthermore, we address generalizability limitations in the Limitations section (Line 371-374): Current arterial occlusion pressure (AOP) recommendations span 40-80%[51], while applied pressures range widely (50-230 mmHg across literature). Our use of standardized 200 mmHg pressure - implemented for safety and operational feasibility in stroke patients, which may impact generalizability.

3. Comments Clarify that short-term strength gains likely reflect neural adaptation rather than hypertrophy.

Response: We sincerely appreciate the reviewer's insightful comment regarding the mechanistic interpretation of short-term strength gains. We fully agree that early-phase improvements in strength are predominantly driven by neural adaptations rather than hypertrophic changes. To explicitly address this point:

Discussion Reinforcement:

As noted in our Discussion (lines 261-322), we have consistently emphasized the role of acute and chronic neural adaptations as the primary mechanism underpinning early strength gains. 

Following the reviewer’s suggestion, we have added the following statement to the Conclusions section (Lines 381-384) to crystallize this distinction The present study highlights the superiority of BFRT and HLRT over LLRT alone for the development of central activation and muscle performance through neural adaptations rather than hypertrophic changes. Training programs should therefore prioritize neuromodulatory stimuli via targeted exercise variable manipulation.

4. Comments Include ITT as sensitivity analysis in future studies or at least discuss this limitation.

Response: We sincerely thank the reviewer for raising this important point regarding the use of intention-to-treat (ITT) analysis. As the reviewer suggested, we have explicitly discussed this limitation in the revised manuscript (Line374-377)�Third, our analytical approach followed per-protocol rather than intention-to-treat (ITT) principles. Future investigations should incorporate ITT analysis to evaluate robustness against attrition bias and maintain randomization integrity.

We confirm that all figure files have been successfully processed through the PACE tool prior to resubmission. 

We tried our best to improve the manuscript and made some changes in the manuscript. We think that our manuscript has greatly improved after revisions, and we would like to submit the revised manuscript for publication in Plos one. We appreciate the reviewers’ warm work earnestly and hope that the correction will meet with approval. Once again, thank you very much for your comments and suggestions.

Best wishes,

July 2th, 2025

---

## [Decision Letter · Decision Letter 2]

23 Sep 2025

Cerebral cortical activation and muscle performance during blood flow restriction training after ischemic stroke: a randomised functional near-infrared spectroscopy study

PONE-D-25-08906R2

Dear Dr. Zhu,

We’re pleased to inform you that your manuscript has been judged scientifically suitable for publication and will be formally accepted for publication once it meets all outstanding technical requirements.

Kind regards,

Miray Budak

Academic Editor

PLOS ONE

Additional Editor Comments (optional):

Reviewer #1:

Reviewer #5:

Reviewers' comments:

Reviewer's Responses to Questions

**Comments to the Author**

Reviewer #1: All comments have been addressed

Reviewer #5: All comments have been addressed

2. Is the manuscript technically sound, and do the data support the conclusions?

Reviewer #1: (No Response)

Reviewer #5: Yes

3. Has the statistical analysis been performed appropriately and rigorously?

Reviewer #1: (No Response)

Reviewer #5: Yes

4. Have the authors made all data underlying the findings in their manuscript fully available?

Reviewer #1: (No Response)

Reviewer #5: Yes

5. Is the manuscript presented in an intelligible fashion and written in standard English?

Reviewer #1: (No Response)

Reviewer #5: Yes

Reviewer #1: (No Response)

Reviewer #5: Thank you for submitting the manuscript titled ‘Cerebral cortical activation and muscle performance during blood flow restriction training after ischaemic stroke: a randomised functional near-infrared spectroscopy study‘. This is a good trial that adds to the current literature.

I believe that this study is strong and that the comments have been addressed. I usually critique each section, but I believe that this draft is good/sufficient enough for publication.

**Do you want your identity to be public for this peer review?** For information about this choice, including consent withdrawal, please see our Privacy Policy

Reviewer #1: No

Reviewer #5: No

---

## [Editor Report · Acceptance letter]

PONE-D-25-08906R2

PLOS ONE

Dear Dr. Zhu,

I'm pleased to inform you that your manuscript has been deemed suitable for publication in PLOS ONE. Congratulations! Your manuscript is now being handed over to our production team.

Kind regards,

on behalf of

Dr. Miray Budak

Academic Editor

PLOS ONE